# Is Synthetic Data Ready for Improving Visual Grounding?

## Abstract

This paper extensively investigates the effectiveness of synthetic training data to improve the capabilities of vision-and-language models for grounding textual descriptions to image regions. We explore various strategies to best generate image-text pairs and image-text-box triplets using a series of pretrained models under different settings and varying degrees of reliance on real data. Through comparative analyses with synthetic, real, and web-crawled data, we identify factors that contribute to performance differences, and propose *SynGround*, an effective pipeline for generating useful synthetic data for visual grounding. Our findings show that SynGround can improve the localization capabilities of off-the-shelf vision-and-language models and offers the potential for infinite data generation. Particularly, SynGround improves the pointing game accuracy of pretrained ALBEF and BLIP models by 4.81% and 17.11% absolute percentage points, respectively, across the RefCOCO+ and the Flickr30k benchmarks.

## 1 Introduction

Vision-and-language models pretrained on large-scale image and text pairs have become exceedingly accurate across various tasks (Lu et al., 2019; Li et al., 2019; Jia et al., 2021; Li et al., 2021; 2022b; Radford et al., 2021; Ma et al., 2023; Bitton-Guetta et al., 2023; Paiss et al., 2023). By leveraging web-sourced datasets, these models showcase a strong ability to comprehend and process an extensive vocabulary of objects and scenes, demonstrating remarkable performance. Our work focuses on the task of visual grounding, which consists of mapping arbitrary input text to image regions. Recent methods finetune vision-and-language models pretrained on web-scale image-text pairs with a large but more modest number of images annotated with bounding boxes or other region annotations; alternatively, these methods leverage pretrained object detectors that have been trained on such annotated data (Chen et al., 2020; Dou & Peng, 2021; Gupta et al., 2020; Yang et al., 2023; Li et al., 2022b; Kamath et al., 2021; Yang et al., 2022; Chen et al., 2023; Jiang et al., 2022). The resulting vision-and-language models can then be used to perform visual grounding over an arbitrary vocabulary of objects.

Collecting annotations for tasks that require localizing objects is considerably more expensive than for other tasks. Region annotations in the form of bounding boxes or segments can not be easily obtained from the web in the same way that image-text pairs can be found, and require more cognitive effort to annotate manually than just providing a textual label. Recent work has advocated for the use of synthetic data – *learning from models* – even for tasks that require only image-text pair supervision (Tian et al., 2023) due to the poor scalability of large-scale uncurated data (Schuhmann et al., 2022). Our work takes this paradigm one step further by investigating whether synthetic data obtained from models is ready to make significant improvements for the visual grounding task, where we need to obtain high-quality samples in the form of image-text-region triplets.

In this paper, we take advantage of recent advancements in text-to-image generation (Nichol et al., 2021; Rombach et al., 2022; Saharia et al., 2022), large language models (Touvron et al., 2023; Chiang et al., 2023) and models for other vision-and-language tasks (Liu et al., 2024; Li et al., 2023a; 2022b) to design an effective pipeline to supervise vision-and-language models for visual grounding. We refer to this pipeline as *SynGround* and present a systematic analysis to justify each stage of our data generation process. While there have been several attempts in training visual recognition models with synthetic data by leveraging automatically generated image-text pairs (He et al., 2022a; Azizi

et al., 2023; Fan et al., 2023; Tian et al., 2023; 2024; Sariyildiz et al., 2023), our work is the first to also leverage generative models for synthesizing grounded image data. Moreover, we assess the efficacy of synthetic data by comparing it to real and web-crawled data, identifying specific factors that limit its performance. We also investigate whether synthetic data can augment real data and examine its scalability.

Our key findings and contributions are summarized as follows: (1) For a text-to-image generative model, detailed prompts obtained from image captioners yield the most effective synthetic image-text pairs for visual grounding, surpassing those generated from concatenated region descriptions or LLM-generated text. (2) To obtain synthetic image-text-boxes, both layout-conditioned generative models and object detectors using synthetic image-text pairs show promise. However, layout-conditioned models are more limited due to the observed non-overlap and natural input layout requirements. (3) We use our findings to propose SynGround, an effective pipeline to generate data for visual grounding through image-text-box synthesis. This method leverages exhaustive image descriptions for image synthesis, an LLM for text synthesis from phrase extraction, and an open-vocabulary object detector for bounding box generation. (4) Our results show that using our generated synthetic data outperforms using web-crawled data (Sec. 3.9). Additionally, our synthetic data can effectively augment real data (Sec. 3.4) and shows an upward trend in terms of scalability (Sec. 3.8).

## 2 RELATED WORK

**Visual Grounding.** Visual grounding associates textual descriptions with relevant regions within images. Supervised methods are typically trained with image-text-box pairs (Deng et al., 2018; 2021; Dou & Peng, 2021; Kamath et al., 2021; Yang et al., 2023), or integrate pretrained object detectors (Ren et al., 2015; He et al., 2017) to identify the most relevant regions with respect to textual descriptions (Chen et al., 2020; Datta et al., 2019; Gomel et al., 2023; Gupta et al., 2020; Lu et al., 2020; Wang & Specia, 2019). While weakly-supervised methods bypass the need for bounding boxes (Arbelle et al., 2021; Shaharabany & Wolf, 2023; Shaharabany et al., 2022; He et al., 2023), they rely on datasets such as Visual Genome (Krishna et al., 2017), which provides multiple phrases describing various regions in each image. However, the process of manually annotating dense textual descriptions and their corresponding boxes is time-consuming. Although some studies collect more data (Xiao et al., 2023) or generate annotations for existing image-text datasets (Peng et al., 2023; You et al., 2023; Wang et al., 2023), we posit that our contribution is orthogonal as we aim to investigate the feasibility and limitations of generating and using synthetic data. Related to grounding methods incorporating tuning of visual explanations (Xiao et al., 2017; Li et al., 2021; Yang et al., 2023; He et al., 2023), we explore visual grounding in a more general context, aiming to localize phrases using gradient-based model explanations(i.e. GradCAM (Selvaraju et al., 2017)) rather than generating boxes (Li et al., 2022b). Compared to boxes, explanation maps provide a more flexible representation that can be used for text referring to multiple objects or background regions. Yang *et al.* (Yang et al., 2023) recently proposed an attention mask consistency objective to optimize the gradient-based explanations of ALBEF (Li et al., 2021) to improve localization performance. We adopt ALBEF as our main base model and tune it with attention mask consistency on image-text-box triplets.

**Learning from Synthetic Data.** The use of synthetic data has been widely explored across various computer vision tasks, including image classification (Gan et al., 2020; Peng et al., 2017; Mishra et al., 2022), semantic segmentation (Richter et al., 2016; Ros et al., 2016; Chen et al., 2019), object detection (Peng et al., 2015; Rozantsev et al., 2015), human pose estimation (Varol et al., 2017; Kim et al., 2022), and many other domains (Abu Alhaija et al., 2018; Varol et al., 2021; Dan et al., 2020; He et al., 2022b; Kumar et al., 2020; Meng et al., 2022; Mimura et al., 2018; Rosenberg et al., 2019; Rossenbach et al., 2020; Tucker et al., 2020; Yang et al., 2020; Moreau et al., 2022; Yen-Chen et al., 2022). In contrast to works that generate synthetic data using 3D-rendering (Greff et al., 2022; Zheng et al., 2020) or physically realistic engines (de Melo et al., 2022; Dosovitskiy et al., 2017; Gan et al., 2020; Cascante-Bonilla et al., 2023; 2022), our approach aligns more closely with research adopting diffusion models. He *et al.* (He et al., 2022a) use GLIDE (Nichol et al., 2021) for generating synthetic images to improve a pretrained CLIP model (Radford et al., 2021) in zero-shot and few-shot classification, while its performance is adversely affected when trained from scratch on synthetic data. Azizi *et al.* (Azizi et al., 2023) fine-tune Imagen (Saharia et al., 2022) on ImageNet (Russakovsky et al., 2015) and subsequently leverage its synthetic data to augment the real ImageNet training set, resulting in initial improvement followed by degradation upon scaling up. Fan *et al.* (Fan et al.,

2023) investigate the scaling laws of synthetic images and identify related factors. StableRep (Tian et al., 2024) propose a self-supervised method with a multi-positive contrastive loss that learns representations from synthetic images generated for captions in large-scale datasets (Changpinyo et al., 2021; Desai et al., 2021), thereby boosting linear probing image classification performance. SynCLR (Tian et al., 2023) uses LLM-generated synthetic captions. Our research not only generates image-text pairs but also provides corresponding synthetic boxes, facilitating a comprehensive exploration of the efficacy of synthetic image-text-box triplets in visual grounding.

## 3 IS SYNTHETIC DATA READY FOR IMPROVING VISUAL GROUNDING?

We investigate effective strategies to generate image-text-boxes $\langle I, T, B \rangle$ to improve the visual grounding ability of a base vision-and-language model. The base model comprises a text encoder $\phi_t$, a visual encoder $\phi_v$, and a multimodal fusion encoder $\phi_f$. Sec. 3.1 introduces the objectives for tuning the base model on image-text pairs $\langle I, T \rangle$ and image-text-box triplets $\langle I, T, B \rangle$. Sec. 3.2 explores various image-text synthesis strategies with an image generation model $\Psi_g$, while Sec. 3.3 delves into multiple approaches for box synthesis. In the following sections, we conduct extensive experiments and analyses with our proposed image-text-box synthesis paradigm, SynGround, which integrates an image caption generator $\Psi_c$, a text-to-image generation model $\Psi_g$, a large language model $\Psi_t$ and an object detector $\Psi_d$. We cover topics including the effect of augmenting real data (Sec. 3.4), factors contributing to performance discrepancies compared to real data (Sec. 3.5), effectiveness and analyses with other VLMs (Sec. 3.6), a more flexible design for generating theoretically infinite data (Sec. 3.7), an analysis on the effect of scale (Sec. 3.8), comparisons with web-crawled data (Sec. 3.9), and implementation details (Sec. 3.10).

### 3.1 PRELIMINARIES AND SETUP

**Image-Text Matching.** We adopt ALBEF (Li et al., 2021) as the main base model which incorporates image-text matching objectives including a standard image-text matching loss ($\mathcal{L}_{\mathrm{itm}}$), an image-text contrastive loss ($\mathcal{L}_{\mathrm{itc}}$) and a masking language modeling loss ($\mathcal{L}_{\mathrm{mlm}}$). The image-text matching loss $\mathcal{L}_{\mathrm{itm}}$ evaluates the compatibility between an image and a text by analyzing the output of [CLS] tokens. This loss measures how well a given image-text pair $\langle I, T \rangle$ matches using a cross-entropy loss. The image-text contrastive loss $\mathcal{L}_{\mathrm{itc}}$ is designed to align visual and textual representations using contrastive learning by sampling a set of negative samples and a temperature scaling parameter to normalize the scores. The masking language modeling loss $\mathcal{L}_{\mathrm{mlm}}$ uses both visual inputs and textual context to predict masked tokens from the input text. The overall objective to tune the base model on image-text pairs is $\mathcal{L}_{\mathrm{vl}} = \mathcal{L}_{\mathrm{itm}} + \mathcal{L}_{\mathrm{itc}} + \mathcal{L}_{\mathrm{mlm}}$.

**Image-Text-Box Matching.** We adopt an attention map consistency objective $\mathcal{L}_{\mathrm{amc}}$, which was recently proposed by Yang *et al.* (Yang et al., 2023) to add region-level box supervision on top of the ALBEF model. This objective uses gradient-based explanation maps $G$ through GradCAM (Selvaraju et al., 2017), and maximizes the consistency between this map and region annotations. This objective considers two terms. The first term $\mathcal{L}_{\mathrm{max}}$ encourages the maximum value of $G$ inside a target box $B$ to surpass the maximum value outside by a margin $\delta_1$.

$$\mathcal{L}_{\mathrm{max}} = \mathbb{E}_{(I,T,B) \sim D} \left[ \max(0, \ \max_{i,j} \left( (1 - B_{i,j}) G_{i,j} \right) - \max_{i,j} \left( B_{i,j} G_{i,j} \right) + \delta_1 \right], \quad (1)$$

where $B_{i,j}$ is 1 when pixel location $i, j$ is inside the box, and zero otherwise. The second term $\mathcal{L}_{\mathrm{mean}}$ encourages the mean value of heatmap $G$ inside the box to be larger than the mean value outside by a margin $\delta_2$.

$$\mathcal{L}_{\mathrm{mean}} = \mathbb{E}_{(I,T,B) \sim D} \left[ \max(0, \ \frac{\sum_{i,j} (1 - B_{i,j}) G_{i,j}}{\sum_{i,j} (1 - B_{i,j})} - \frac{\sum_{i,j} B_{i,j} G_{i,j}}{\sum_{i,j} (B_{i,j})} + \delta_2 \right]. \quad (2)$$

The full $\mathcal{L}_{\mathrm{amc}}$ objective is $\mathcal{L}_{\mathrm{amc}} = \lambda_1 \cdot \mathcal{L}_{\mathrm{max}} + \lambda_2 \cdot \mathcal{L}_{\mathrm{mean}}$, where $\lambda_1, \lambda_2$ are trade-off hyperparameters. The base model is tuned with both the $\mathcal{L}_{\mathrm{vl}}$ and $\mathcal{L}_{\mathrm{amc}}$ objectives on image-text-box triplets.

**Visual Grounding Evaluation.** Following prior works (Yang et al., 2023; Akbari et al., 2019; Li et al., 2021; He et al., 2023; Datta et al., 2019; Lu et al., 2020; Gupta et al., 2020; Dou & Peng, 2021), our evaluation uses pointing game accuracy, which measures the proportion of instances where the maximal activation point within generated heatmaps correctly falls within the annotated ground-truth box regions. We conduct evaluation across multiple benchmarks, including RefCOCO+ (Yu et al., 2016) and Flickr30k (Plummer et al., 2015).

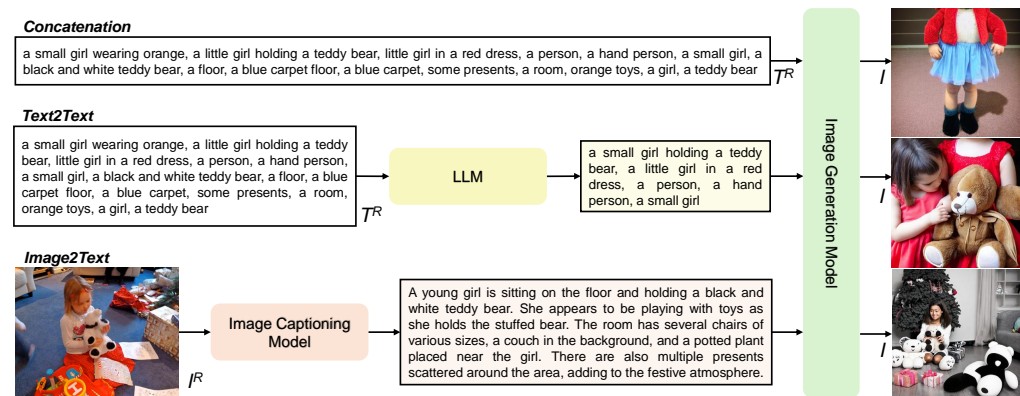

Figure 1: Illustration of various approaches for image and image description synthesis. Image descriptions can be generated by concatenating real text $T^R$, LLM summary on real text $T^R$, and image captioning on real image $I^R$. Synthetic images $I$ are obtained through an image generator model conditioned on image descriptions.

Table 1: Comparisons of image-text synthesis strategies. We assess the effectiveness of synthetic image-text pairs from text concatenation, Text2Text, and Image2Text pipelines, by evaluating the performance improvements over an ALBEF model. For reference we also include the performance that would be obtained by finetuning ALBEF on real image-text pairs from Visual Genome (VG).

| Category | Row | Image | Text | Num. | RefCOCO+ | | Flickr30k | $\Delta_{avg}$ |
|---|---|---|---|---|---|---|---|---|
| | | | | | Test A | Test B | | |
| ALBEF | 1 | - | - | – | 69.35 | 53.77 | 79.38 | - |
| ALBEF + VG | 2 | VG | VG | 1,649,546 | 71.41 | 54.06 | 79.90 | +0.96 |
| Concatenation | 3 | Syn-C | VG | 1,649,546 | 67.57 | 53.14 | 76.99 | -1.60 |
| Text2Text | 4 | Syn-V | VG | 1,649,546 | 67.41 | 52.14 | 77.80 | -1.72 |
| | 5 | Syn-V | LLM$_C$ | 530,233 | 70.28 | 52.08 | 78.97 | -0.39 |
| Image2Text | 6 | Syn-B | VG | 1,649,546 | 56.88 | 48.48 | 73.93 | -7.74 |
| | 7 | Syn-B | BLIP-2$_C$ | 267,199 | 68.15 | 51.50 | 78.30 | -1.52 |
| | 8 | Syn-L | VG | 1,649,546 | 65.35 | 50.28 | 76.85 | -3.34 |
| | 9 | Syn-L | LLaVA$_P$ | 384,455 | 70.22 | 52.30 | 78.34 | -0.55 |
| | 10 | Syn-L | LLaVA$_C$ | 716,198 | 69.94 | 53.26 | 78.83 | -0.16 |
| | 11 | Syn-L | LLaVA$_L$ | 680,093 | 69.84 | **53.61** | 79.44 | +0.13 |
| | 12 | Syn-L | LLaVA$_S$ | 1,031,521 | **70.31** | 52.55 | **80.73** | **+0.36** |

## 3.2 IMPROVING VISUAL GROUNDING USING ONLY SYNTHETIC IMAGE-TEXT PAIRS

To generate image-text-box triplets for visual grounding, we first explore synthesizing image-text pairs that are not only aligned but also inherently equipped for visual grounding. As illustrated in Fig. 1, we investigate three alternatives for conditioning a text-to-image generation model $\Psi_g$. (1) *Concatenation*: merging all captions of a real image $I$ as a prompt for $\Psi_g$. (2) *Text2Text*: Using an LLM $\Psi_t$ to create a cohesive prompt given a set of text descriptions. (3) *Image2Text*: Employing an image captioning model $\Psi_c$ to generate new captions for real images $I^R$ as prompts for $\Psi_g$. Table 1 compares these strategies. We tune all of the models in these experiments using the image-text matching objectives described in Sec. 3.1. Although image-text matching objectives are not designed specifically for visual grounding, well-aligned region phrases from the Visual Genome (VG) dataset can improve the visual grounding performance by 0.96% on average (row 2).

The *Concatenation* strategy (Syn-C) degrades the average performance by 1.60%, indicating that the text-to-image generation model $\Psi_g$ is not effective with long yet potentially redundant prompts. For *Text2Text*, LLM summaries generated synthetic image (Syn-V) show misalignment with the original VG captions (row 4). Also, tuning the model on Syn-V and object-centric phrases obtained by splitting the LLM summary with commas (LLM$_C$) is ineffective (row 5). For the *Image2Text* strategy, we experiment with two distinct styles of image captioning models: BLIP-2 (Li et al.,

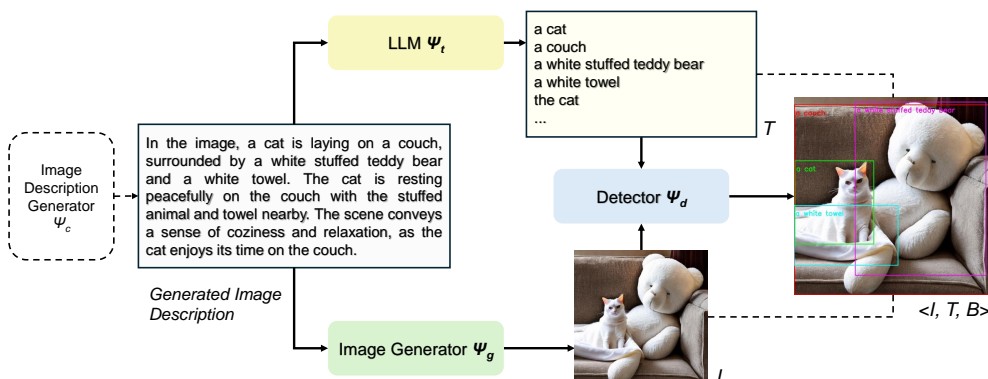

Figure 2: Overview of our image-text-box synthesis pipeline, SynGround. We use an image description generator $\Psi_c$ to output a description that serves as a prompt to an image generator $\Psi_g$ to obtain synthetic image $I$. This description is also used to obtain text phrases $T$ by prompting an LLM $\Psi_t$. Finally, the synthetic text and image are fed into an object detector $\Psi_d$ to obtain synthetic boxes $B$.

2023a) that yields condensed phrases, and LLaVA (Liu et al., 2024) that produces detailed paragraphs. Both BLIP-2 and LLaVA prompted images (Syn-B and Syn-L, respectively) show partial overlap with real VG captions (rows 6 and 8). Notably, an opposite influence is observed when Syn-B and Syn-L are paired with phrases extracted from their captions. BLIP-2 captions, usually short and object-centric (*e.g.,* "a dog, a cat"), are split into visual grounding phrases by commas, showing improved performance over Syn-B and VG captions, possibly due to better cross-modal alignment, but still below the baseline.

We find that LLaVA-synthesized images (Syn-L) paired with phrases extracted from LLaVA captions can enhance grounding performance (Table 1, rows 11 and 12). This indicates that detailed prompts suit the text-to-image synthesis model better. We compare four ways to partition the LLaVA captions into phrases: LLaVA$_P$ and LLaVA$_C$, segmented by periods and commas, respectively, LLaVA$_L$ for longer LLM extracted phrases and LLaVA$_S$ for shorter phrases. Our experiments demonstrate that the *Image2Text* strategy, particularly with LLaVA captioning and LLM phrase extraction, yields the most effective synthetic image-text pairs for visual grounding. *More details in Appendix A.2 and A.3.*

### 3.3 IMPROVING VISUAL GROUNDING WITH SYNTHETIC IMAGE-TEXT-BOX TRIPLETS

This section discusses two pipelines for image-text-box synthesis. The first pipeline builds on the success of *Image2Text* (Sec. 3.2) and additionally uses an open vocabulary object detector $\Psi_d$(Li et al., 2022b) to generate region annotations for each synthetic text phrase. Fig. 2 shows an overview of this strategy. As shown in Table 2, we compare pairing the synthetic images with shorter phrases (LLaVA$_S$), longer phrases (LLaVA$_L$), and both (LLaVA$_{S,L}$). The shorter phrases outperform others (row 10), leading to an average performance gain of 4.81%. However, combining shorter and longer phrases (LLaVA$_{S,L}$) –despite increasing the amount of data– does not further improve performance, suggesting redundancy in the information conveyed by phrases with different lengths.

We also investigate an alternative strategy that leverages a layout-conditioned generative model GLIGEN (Li et al., 2023b), synthesizing images conditioned on the text and corresponding bounding boxes. Directly inputting all real VG texts and boxes (row 3) results in a modest increase of 2.58% compared to the baseline (row 1). We observe the ineffectiveness of using regions with multiple textual descriptions, as this tends to generate unrealistic or implausible content. To address it, we explore three strategies: Random selection of text-box inputs (VG$_R$), reduction based on average CLIP (Radford et al., 2021) text dissimilarity (VG$_T$), and selecting the maximum number of boxes with an IoU below 0.5 (VG$_I$). Random selection keeps at most 10 boxes per image, resulting in a reduction of about 50% of the data. Random text-box synthesized images Syn-R (row 5) outperform the all-text-box conditioned variant (Syn-A, row 3). Also, pairing Syn-R with all text-box data from Real VG (row 4) does not match the effectiveness of either Syn-A with all text-boxes or Syn-R with selected text-boxes, underscoring the importance of image-text-box alignment. Sorting by CLIP text dissimilarity to select at most top-10 inputs (Syn-T, VG$_T$) marginally improves the random selection.

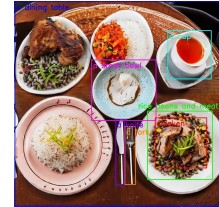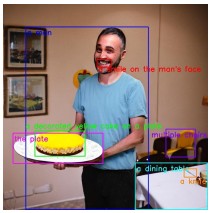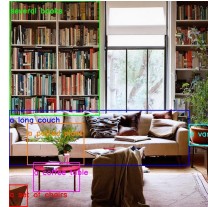

Figure 3: Qualitative examples of synthetic image-text-box triplets from SynGround.

Table 2: Effectiveness of synthetic image-text-boxes generated with either GLIP (Li et al., 2022b) or GLIGEN (Li et al., 2023b). For reference we also include the reported performance obtained by finetuning ALBEF (Li et al., 2021) with an AMC loss (Yang et al., 2023) on real image-text-box triplets from Visual Genome (VG).

| Model | Row | Image | Text | Box | Num. | RefCOCO+ | | Flickr30k | $\Delta_{avg}$ |
|---|---|---|---|---|---|---|---|---|---|
| | | | | | | Test A | Test B | | |
| ALBEF | 1 | - | - | - | – | 69.35 | 53.77 | 79.38 | - |
| AMC | 2 | VG | VG | VG | 1,649,546 | 78.89 | 61.16 | 86.46 | +8.00 |
| ALBEF | 3 | Syn-A | VG | VG | 1,649,546 | 68.79 | 56.88 | 84.57 | +2.58 |
| | 4 | Syn-R | VG | VG | 1,649,546 | 68.25 | 55.78 | 84.59 | +2.04 |
| + | 5 | Syn-R | $VG_R$ | $VG_R$ | 725,974 | 71.66 | 56.15 | 84.84 | +3.38 |
| | 6 | Syn-T | $VG_T$ | $VG_T$ | 725,974 | 71.80 | 56.68 | 84.73 | +3.57 |
| GLIGEN | 7 | Syn-I | $VG_I$ | $VG_I$ | 652,657 | 73.05 | 58.38 | 84.39 | +4.44 |
| ALBEF | 8 | Syn-L | $LLaVA_L$ | GLIP | 659,927 | 72.39 | 55.94 | 86.53 | +4.12 |
| + | 9 | Syn-L | $LLaVA_{S,L}$ | GLIP | 1,658,333 | 72.25 | **57.05** | 86.71 | +4.50 |
| GLIP | 10 | Syn-L | $LLaVA_S$ | GLIP | 998,406 | **73.70** | 56.35 | **86.89** | **+4.81** |

Yet, the most significant improvement stems from selecting as many boxes as possible with an IoU below 0.5. The images (Syn-I) generated with this strategy match the best practice in the GLIP-based pipeline (row 10).

Our results show the potential of using a layout-conditioned generative model for image-text-box synthesis. However, either generating non-overlapping and natural layouts or generating text for visually coherent layouts poses a substantial challenge, limiting the advancement to synthesis without real image-text-box data. Even with layout generation models (Inoue et al., 2023; Kikuchi et al., 2021), strong constraints of natural composition and non-overlapping bounding boxes detract from their efficiency and effectiveness compared to the object detector approach.

We use our findings to define **SynGround**, a processing pipeline for generating synthetic image-text-boxes for visual grounding (Table 2 row 10, Fig. 2). Fig. 3 shows representative examples of our generated image-text-boxes, including images with specific and recognizable entities (the first image shows "a Siamese cat"), complex scenarios with composite subjects (the second image shows "rice, beans and meat"). The third image shows a synthetic person with unrealistic features, observed in several generated results. This contrasts with improvements on RefCOCO+ Test A (a person-only subset), suggesting that realistic object details are not crucial for visual grounding. The fourth image showcases creative objects with unusual attributes such as a pink coffee table, which showcases diversity in our generated data. *More qualitative examples are provided in Appendix F.*

## 3.4 IMPROVING VISUAL GROUNDING USING BOTH REAL AND SYNTHETIC DATA

SynGround can augment training with real data. Table 3 presents comparisons between training exclusively on real data from the Visual Genome (VG) dataset, synthetic data from SynGround, and a combination of both. The baseline performance (row 1) is significantly enhanced by incorporating synthetic data, yielding an average improvement of 4.81% (row 3). While it falls short of the gains achieved through training on real data (row 2), SynGround offers an average improvement of 9.16% when combined with real data (row 4), outperforming the state-of-the-art (row 2) (Yang et al., 2023) on RefCOCO+ (Yu et al., 2016) Test A and B, and Flickr30k (Plummer et al., 2015) benchmarks.

Table 3: Training on both synthetic and real data. We compare visual grounding improvements for the base model (row 1) by using the real data (row 2), synthetic data (row 3), and both (row 4).

| Method | Data | Num. | RefCOCO+ | | Flickr30k | $\Delta_{avg}$ |
|---|---|---|---|---|---|---|
| | | | Test A | Test B | | |
| ALBEF (Li et al., 2021) | Off-the-Shelf | – | 69.35 | 53.77 | 79.38 | - |
| AMC (Yang et al., 2023) | Real | 1,649,546 | 78.89 | 61.16 | 86.46 | +8.00 |
| SynGround$_S$ | Synthetic | 998,406 | 73.70 | 56.35 | 86.89 | +4.81 |
| SynGround | Real&Synthetic | 2,627,952 | **79.06** | **63.67** | **87.26** | **+9.16** |

Table 4: Factors causing the performance gap with the real data. We investigate how each model caused the ineffectiveness compared to the real data. I: Off-the-shelf base model. II: Learning from real data. III-V: Sequentially replacing real boxes, text, and images with synthetic variants.

| Exp. | Image | Text | Box | Num. | RefCOCO+ | | Flickr30k | $\Delta_{avg}$ |
|---|---|---|---|---|---|---|---|---|
| | | | | | Test A | Test B | | |
| I | - | - | - | – | 69.35 | 53.77 | 79.38 | - |
| II | VG | VG | VG | 1,649,546 | 78.89 | 61.16 | 86.46 | +8.00 |
| III | VG | VG | GLIP | 1,599,633 | 76.88 | 59.79 | 86.76 | +6.98 |
| IV | VG | LLaVA$_S$ | GLIP | 1,000,634 | 73.11 | 57.35 | 87.49 | +5.15 |
| V | Syn-L | LLaVA$_S$ | GLIP | 998,406 | 73.70 | 56.35 | 86.89 | +4.81 |

## 3.5 FROM REAL DATA TO SYNTHETIC DATA: PERFORMANCE GAP FACTORS

Table 4 analyzes the factors contributing to the performance gap between synthetic and real data. Experiment I is the off-the-shelf ALBEF performance, serving as a baseline. Experiment II provides the results from training on real VG image-text-boxes, leading to an average improvement of 8%. Experiment III retains real images and texts from VG, but employs GLIP-generated boxes. The 1.02% decrease in performance compared to Experiment II suggests that the synthetic boxes, while effective, may lack the precision of manual-annotated equivalents. Experiment IV further replaces real VG captions with synthetic captions from SynGround (*i.e.,* LLaVA$_S$), resulting in an additional average reduction of 1.83%. This decline could stem from a reduction in the number of captions (∼600K fewer) or discrepancies in image-text alignment, coverage, and diversity compared to manually curated captions (*details in Appendix D*). Interestingly, the performance on Flickr30k is enhanced by 1.03% over real data (II), showing a potential distribution shift from synthetic captions. In Experiment V, the setting consists entirely of synthetic image-text-box data, eliminating real images from the dataset. Compared to Experiment IV, it modestly drops another 0.34%. This minor decrement, relative to the changes observed with synthetic texts and boxes, indicates that synthetic images maintain a level of effectiveness for visual grounding tasks comparable to their real counterparts.

## 3.6 EFFECTIVENESS AND GENERALIZATION ON OTHER VLMS

This section experiments with an additional off-the-shelf VLM, BLIP (Li et al., 2022a), to further examine the effectiveness of our synthetic data and verify the generalizability of our findings from the default base model ALBEF. *Refer to Appendix. B for the base model selection and implementation details.* As shown in Table 5, our generated synthetic image-text and image-text-boxes significantly enhance its visual grounding performance (III, V), matching closely to the improvement from training on real data (II, IV). Additionally, we investigate the factors contributing to the degradation in Table 6. Similar to findings from ALBEF experiments in Table 3.5, most drop comes from the box and text synthesis. By replacing ALBEF with BLIP for experiments presented in previous sections, consistent findings are observed (*details in Appendix C*).

## 3.7 EFFECT OF LESS TO NO RELIANCE ON REAL IMAGES

In this section, we explore a series of variants of our methodology that we refer to as SynGround$^H$, which consists of synthesizing image-text-boxes with less or even no reliance on real data. SynGround$_S^H$ substitutes real images and the image captioning model with an extracted concept list,

Table 5: Training on both synthetic and real data. We compare visual grounding improvements for BLIP (I) by using the real data (II, IV) and synthetic data (III, V) w/o the AMC box-supervised loss.

| Exp. | Box (AMC Loss) | Data | Num. | RefCOCO+ | | Flickr30k | $\Delta_{avg}$ |
| --- | --- | --- | --- | --- | --- | --- | --- |
| | | | | Test A | Test B | | |
| I | × | Off-the-Shelf | – | 58.56 | 38.00 | 64.54 | - |
| II | × | Real | 1,649,546 | 68.86 | 52.85 | 64.08 | +8.23 |
| III | × | Synthetic | 998,406 | 63.45 | 44.39 | 68.21 | +4.98 |
| IV | ✓ | Real | 1,649,546 | 78.47 | 61.96 | 85.35 | +21.56 |
| V | ✓ | Synthetic | 998,406 | 71.78 | 54.82 | 85.83 | +17.11 |

Table 6: Performance gap between real and synthetic data analyses with BLIP. We investigate how each model caused the ineffectiveness compared to the real data. I: Off-the-shelf. II: Trained on real data. III-V: Sequentially replacing real boxes, text, and images with synthetic variants.

| Exp. | Image | Text | Box | Num. | RefCOCO+ | | Flickr30k | $\Delta_{avg}$ |
| --- | --- | --- | --- | --- | --- | --- | --- | --- |
| | | | | | Test A | Test B | | |
| I | - | - | - | – | 58.56 | 38.00 | 64.54 | - |
| II | VG | VG | VG | 1,649,546 | 78.47 | 61.96 | 85.35 | +21.56 |
| III | VG | VG | GLIP | 1,599,633 | 75.72 | 58.50 | 86.11 | +19.74 |
| IV | VG | LLaVA$_S$ | GLIP | 1,000,634 | 72.44 | 55.94 | 86.72 | +18.00 |
| V | Syn-L | LLaVA$_S$ | GLIP | 998,406 | 71.78 | 54.82 | 85.83 | +17.11 |

an in-context learning example database, and a large language model (LLM). Fig. 4 presents an overview of SynGround$_S^H$.

The *Image2Text* strategy, detailed in Section 3.3, applies an image captioning model to obtain detailed descriptions from a real image $I^R$ (II). In contrast, *Concept2Text* reduces real data dependency by sampling from a predefined concept list and an in-context learning example database of detailed captions. The concept list is collected from real text $T^R$, and the in-context learning example database is built through image captioning on a small subset of real images $I^R$ (III-V), web-crawled images (VI), or manual-crafted descriptions (VII). Leveraging the in-context learning capability of an LLM $\Psi_t$, *Concept2Text* can theoretically generate unlimited data.

As shown in Table 7, though relying on less or even no real data, the *Concept2Text* strategies (SynGround$_S^H$) not only rivals but match the performance of the *Image2Text* variant on benchmarks. Sourcing in-context examples (ICE) from captioning on real images, web-crawled data, or manual-crafted text descriptions, while reducing the reliance on real data, all achieve absolute average improvements of around 4%. It indicates the potential of generating synthetic data in a more scalable and flexible setting. *Refer to Appendix A.1 for more implementation details.*

## 3.8 EFFECT OF DATA SCALE ON VISUAL GROUNDING

This section explores the potential for scaling synthetic data. We analyze how the performance of SynGround scales by using one fourth, half, and seventy five percent of our total generated almost 1 million image-text-box triplets. We perform experiments 3 times for each scale to measure variance. Fig. 5 illustrates the average pointing game accuracy improvement across RefCOCO+ (Yu et al., 2016) and Flickr30k (Plummer et al., 2015). We plot the mean improvement at each scale with lines and their standard deviations with error bars. The observed upward trend indicates a promising scaling-up ability of using synthetic data with SynGround.

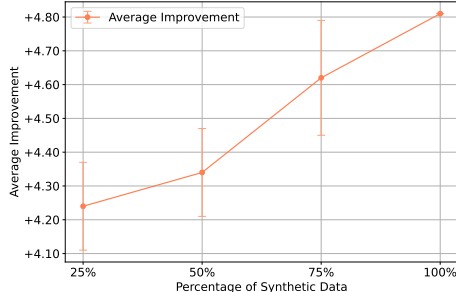

Figure 5: Pointing game accuracy improvement on RefCOCO+ and Flickr30k at various scales. The line denotes the mean improvement across 3 sampled subsets at each scale, and the error bars are corresponding standard deviations.

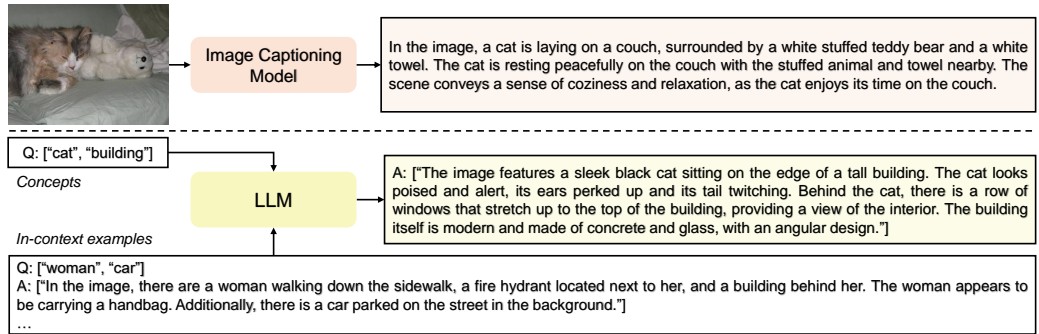

Figure 4: Two approaches for generating image descriptions ($\Psi_C$) for image synthesis and phrase extraction. The top pipeline, *Image2Text*, relies more on real data, applying an image captioning model to real images. The bottom pipeline, *Context2Text*, samples concepts from a predefined list and uses an LLM with in-context learning to generate image descriptions.

Table 7: Performance comparisons with pipelines at different real data reliance due to different image description generators. SynGround$_S$ relies more on real data, whereas SynGround$_S^H$ reduces reliance through a concept list and in-context examples from different sources.

| Category | Exp. | ICE | VG Img. | Source | RefCOCO+ | | Flickr30k | $\Delta_{avg}$ |
|---|---|---|---|---|---|---|---|---|
| | | | | | Test A | Test B | | |
| Off-the-Shelf | I | - | - | - | 69.35 | 53.77 | 79.38 | - |
| SynGround$_S$ | II | - | 94,893 | Real | 73.70 | 56.35 | 86.89 | +4.81 |
| | III | 50 | 50 | Real | 72.48 | 56.23 | 86.07 | +4.09 |
| | IV | 100 | 100 | Real | 72.49 | 56.25 | 86.33 | +4.19 |
| SynGround$_S^H$ | V | 500 | 500 | Real | 72.18 | 55.92 | 86.30 | +3.97 |
| | VI | 500 | 0 | Web-Crawled | 72.69 | 55.66 | 86.29 | +4.05 |
| | VII | 500 | 0 | Manual-Crafted | 71.27 | 56.82 | 86.78 | +4.12 |

## 3.9 COMPARING THE USE OF SYNTHETIC DATA VS. WEB-CRAWLED DATA

To showcase the challenge and necessity of generating effective synthetic data tailored for visual grounding, Table 8 compares our synthetic data and web-crawled data. The first and second rows are the off-the-shelf and tuning on real VG data, respectively. For fair comparisons, we randomly sample 1M web-crawled data from Conceptual Captions (CC) (Sharma et al., 2018), approximately matching the scale of our synthetic data. As CC data only encompasses images and texts, we add synthetic boxes using an open-vocabulary detector (Li et al., 2022b), as the same in our method. Tuning the base model on it achieves (row 3) a 1.82% average performance gain. Additionally, experiments in Table 1 and other work (He et al., 2023) find that visual grounding ability can be enhanced more significantly with object-centric short phrases rather than generic image descriptions. Considering that CC text might describe entire scenarios, we further apply our LLM phrase extraction (row 4) and generate synthetic boxes for the synthetic text phrases, leading to a greater average improvement of 2.86%. However, to our best effort, we can not make the web-crawled data reach a similar enhancement with our synthetic data (SynGround$_S^H$, SynGround$_S$). Our experimental results indicate that it is non-trivial to curate or synthesize image-text-boxes for visual grounding. The image and text favored by visual grounding seem to feature specific properties, such as images with multiple objects and text for region descriptions.

## 3.10 IMPLEMENTATION DETAILS

**Image-Text-Box Synthesis.** To favor reproducibility and accessibility, we adopt Stable Diffusion 2.1 (Rombach et al., 2022) with guidance scale 10.0 as the text-to-image generator $\Psi_g$, an open-source LLM Vicuna-13B (Chiang et al., 2023) as $\Psi_t$, and GLIP (Li et al., 2022b) as the object detector $\Psi_d$. We select the box with top-1 confidence if it exceeds the default confidence threshold (0.7) as in the official implementation. For image description generation $\Psi_c$, we experiment with BLIP-2 (Li et al., 2023a) and LLaVA (Liu et al., 2024) for the *Image2Text* strategy. For the *Context2Text* variant, we

Table 8: Comparisons of our synthetic data with web-crawled data. The first row is the off-the-shelf base model performance, and the second is the performance after tuning on real data. The third row ("CC") tunes on a subset of CC (Sharma et al., 2018) image-text pairs with generated synthetic boxes, while "CC$_{Phrase}$" processes the text through LLM phrase extraction. SynGround$_S^H$ and SynGround$_S$ refer to tuning on our synthetic data, relying on less or more on the real data during synthesis, respectively.

| Method | Data | Num. | RefCOCO+ | | Flickr30k | $\Delta_{avg}$ |
| --- | --- | --- | --- | --- | --- | --- |
| | | | Test A | Test B | | |
| ALBEF (Li et al., 2021) | - | – | 69.35 | 53.77 | 79.38 | - |
| AMC (Yang et al., 2023) | Real | 1,649,546 | 78.89 | 61.16 | 86.46 | +8.00 |
| CC | Web-Crawled | 1,000,000 | 69.05 | 54.96 | 83.94 | +1.82 |
| CC$_{Phrase}$ | Web-Crawled | 1,000,000 | 70.35 | 55.31 | 85.43 | +2.86 |
| SynGround$_S^H$ | Synthetic | 719,254 | 71.27 | 56.82 | 86.78 | +4.12 |
| SynGround$_S$ | Synthetic | 998,406 | **73.70** | **56.35** | **86.89** | **+4.81** |

use Vicuna-13B (Chiang et al., 2023) to generate image descriptions from a two-concept query with four randomly sampled in-context examples. The concept list contains nouns extracted from real VG captions. The in-context learning example database implementation details are in Appendix A.1.

**Visual Grounding Tuning.** We employ ALBEF-14M (Li et al., 2021) as our base model for its reported visual grounding performance through GradCAM (Selvaraju et al., 2017). ALBEF is pretrained on image-text pairs from Conceptual Captions (Changpinyo et al., 2021), ImageNet-1k (Russakovsky et al., 2015), MS-COCO (Lin et al., 2014), SBU Captions (Ordonez et al., 2011) and Visual Genome (Krishna et al., 2017). Tuning for visual grounding applies $\mathcal{L}_{vl}$ on image-text pairs and a combination of $\mathcal{L}_{vl}$ and $\mathcal{L}_{amc}$ on image-text-box triplets, adhering to the coefficient settings $\delta_1 = 0.5$, $\delta_2 = 0.1$, $\lambda_1 = 0.8$, and $\lambda_2 = 0.2$ as originally proposed in Yang *et al.* (Yang et al., 2023). The training is conducted on a single node with 8 NVIDIA A40 GPUs. Input images are resized to 256×256 pixels and augmented with color jittering, horizontal flipping, and random grayscale conversion. All ALBEF-based experiments use an Adam optimizer (Kingma & Ba, 2014) with a learning rate set to 1e-5 and a batch size of 512.

## 4 CONCLUSION

This paper investigates various strategies and conducts extensive analyses for generating synthetic training data to improve the visual grounding ability of a base vision-and-language model. By leveraging exhaustive image descriptions for image synthesis, utilizing an LLM for phrase extraction, and adopting an open-vocabulary object detector for box synthesis, we propose SynGround– an effective framework to generate training data for improving visual grounding. SynGround can augment real data to yield further performance gains, and surpasses the efficacy of web-crawled data in visual grounding. Furthermore, SynGround is scalable and capable of generating theoretically infinite data using LLMs for image description generation.

**Limitations and Future Work.** While SynGround learns from a suite of large-scale pretrained models, it also inherits their limitations, resulting in certain degradations compared to real data. Future improvements could stem from integrating more advanced models, such as GPT-4 (OpenAI, 2023) or DALLE-3 (Betker et al., 2023). Additionally, considering the success and efficiency of SynGround, this work has not yet explored the integration of layout-conditioned image synthesis models with less real-data reliance. Although the proposed *Context2Image* paradigm can theoretically generate unlimited data, practical limitations in computational resources limit our ability to generate and train on larger-scale data. Future studies should investigate the scaling laws applicable under reduced real data reliance.

**Broader Impact.** Using synthetic data for training mitigates privacy issues associated with real images, as the identities of real people are unlikely to be depicted. However, training on synthetic data raises ethical concerns, especially regarding the amplification of implicit biases present in the source data used to train the adopted pretrained models. Such biases may manifest in the oversampling of specific skin colors and genders, such as in certain caption descriptions.

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

In this appendix, we provide additional implementation details in Section A, justification of base model selection in Section B, generalizability of findings in Section C, analyses of synthetic text in Section D, comparisons between alternative paradigm designs in Section E, and more qualitative examples in Section F.

## A    IMPLEMENTATION DETAILS

This section presents implementation details, including concept list sampling, as well as LLM prompts used for generating image descriptions, summarizing captions, and extracting text phrases.

### A.1    CONCEPT2TEXT: CONCEPT LIST AND IN-CONTEXT EXAMPLES

Following previous work (Tian et al., 2023), we assume access to a list of concepts and their distribution in real text $T^R$ (captions from the VG dataset (Krishna et al., 2017)). The concept list curation involves tokenizing the real text $T^R$ and identifying nouns by their part-of-speech (POS) tags. To ensure keeping the relevant information provided in a query for image description generation, we exclude a predefined set of nouns commonly used in prompts or spatial positions: "scene", "scenery", "view", "picture", "image", "photo", "left", "right", "back", "front", "top", "bottom", "middle", "center", "side", "background", "frontmost", "leftmost", "rightmost".

We sample two concepts per query for each image description generation by their frequency in real text $T^R$. As shown in Fig. 6, an in-context learning example consists of a two-noun query ("Q") and an image description ("A"). Relying less on the real images, the image descriptions are generated by an image captioning model (Liu et al., 2024) on 50, 100, or 500 randomly sampled real images $I^R$. In the setting with non-reliance on the real image, we randomly sampled 500 web-crawled images from CC3M (Changpinyo et al., 2021) or manually crafted 500 text descriptions. Then we use POS to randomly extract two nouns as their query. The *Caption2Text* image description generation uses four random in-context learning examples with a random two-noun query.

> **Q:** ["**cup**", "**table**"]
>
> **A:** ["The image showcases a wooden dining table with various food items and tableware arranged on it. A teapot is positioned on the table, along with a kettle and a **cup**. Three donuts are also present on the **table**, with one donut placed near the center, another on the left side, and the third on the right side."]
>
> **Q:** ["**couch**", "**video**"]
>
> **A:** ["In the image, there are two people playing a **video** game together. A woman is focused on the game, holding a video game remote in her hand, while a man stands behind her, watching her play. There is also a **couch** in the room, providing a comfortable seating area for the two individuals. Additionally, there is a clock on the wall."]
>
> **Q:** ["**trees**", "**clock**"]
>
> **A:** ["The image features a tall, multi-story building with a prominent **clock** tower at the top. The clock is large and can be seen in the center of the tower. The building is surrounded by **trees** and has a large clock on one of its sides, visible towards the left side of the structure. The overall scene showcases a clock tower that stands out among the surrounding environment."]
>
> **Q:** ["**chair**", "**vase**"]
>
> **A:** ["The image features a clean, well-furnished apartment with two couches, a **chair**, and a coffee table. The living room area is bright and open, and a large window allows natural light to fill the space. On the coffee table, there is a **vase** with a potted plant, and a book is placed on one of the couches."]
>
> **Q:** ["**cars**", "**trucks**"]
>
> **A:** ["In this scene, a group of people is gathered at a busy street corner with several vehicles, including **cars** and **trucks**, surrounding the area. There are multiple police cars parked on the side of the street, as well as other cars and trucks scattered throughout the scene. Two people are holding cell phones, likely communicating with others or checking for updates."]

Figure 6: Random examples from the in-context learning database. The query "Q" contains two nouns, while the expected answer "A" is a crafted image description incorporating the queried nouns.

### A.2    TEXT2TEXT: LLM SUMMARY

In the *Text2Text* strategy, we prompt an LLM (Chiang et al., 2023) to condense the potentially redundant VG captions $T^R$ for the same image $I^R$ into a summarized version. We include four examples in our prompts, as detailed in Fig. 7. Note that, for each query "Q" and expected answer

**Q:** ["blue metal bike on ground, brown grass fence panel, bicycle parked against the wall, a blue bicycle parked at the railing, a railing of metal pipes and rods, straw fence along the railing, a wood and tan surf board, a tan wood, a red blue and yellow surfboard, a blue and metal bike, an old and rusty bike, a wall, a brown fence panel, a blue surfboard, a surfboard, a white surfboard, a writing, a fence, a boggy board, flame stripes, a brown and boggy board, a blue bicycle, a bicycle, a tan and blue surf board, a railing, a wooden brown and vertical surfboard, metal pipes, a straw fence, a balcony"]

**A:** ["blue metal old rusty bike, a wooden brown and vertical surfboard, a red blue and yellow surfboard, a tan and blue surfboard, straw fence along the railing, brown and boggy board, flame stripes, balcony, a railing of metal pipes and rods"]

**Q:** ["this is a person, the cat is black and white, black cat laying on beige carpet, girl playing with cat, fireplace behind the girl, a screen in front of the fireplace, a here person, a here and black cat, a grey and light ground, a girl, a stone fireplace"]

**A:** ["a girl playing with cat, a black and white cat laying on beige carpet, a grey and light ground, a stone fireplace, a screen in front of the fireplace"]

**Q:** ["men riding horses on set, two men riding horses on a cobblestone road, two horses have riders but two do not, it is a daytime scene, it is an outdoor scene, there are several people in the photo, there are several horses in the photo, a large brown building, two men on horseback, 2 men and 4 horses, 3 men looking at a motorcycle, 2 men ponying 2 horses, telephone poles on village street, a total of eight people are in this picture, four brown horses with white markings, a man, a horse, some men, some horses, some riders, a daytime, a scene, an outdoor, a sunny, some seems, a sky, some people, a photo, a motorcycle, a motoryole, some whites, a top, a cowboy, a brown, a cement, a building, a rider, a parked, a street, a road, a horseback, a rocky, a motorbike"]

**A:** ["two men riding horses on a cobblestone road, four brown horses with white markings, eight people, a large brown building, a motorcycle, telephone poles on village street"]

**Q:** ["a scene in an old building, a window with green trim, a tan painted brick wall, an a, a window, a top"]

**A:** ["an old building, a window with green trim, a tan painted brick wall"]

Figure 7: LLM prompts that summarizes real captions in *Text2Text* strategy. Each example comprises a query "Q" in orange and its expected answer "A" in yellow. "Q" is concatenated real text for an image, and "A" is our crafted summary.

"A", all orange rows correspond to images generated using the captions from the VG dataset ("Q"); as a counterpart, the yellow rows show images generated by the summarized captions ("A"). Compared to directly concatenating all VG captions, the images generated for summary tend to include more salient objects in the prompts (*e.g.,* surfboards in the 2nd row and the girl in the 4th row), enrich the contextual details (*e.g.,* poles in the 6th row), and remain effective for originally concise captions (*e.g.,* the window and wall in the 8th row).

**Q:** ["there are several cars parked on the street, one of which is a red car near the crosswalk"]

**A:** ["several cars", "the street", "a red car", "the crosswalk"]

**Q:** ["on the countertop, there is a white plate and a bowl, two cups, a spoon, and a bottle"]

**A:** ["the countertop", "a white plate", "a bowl", "two cups", "a spoon", "a bottle"]

**Q:** ["the image features a cluttered home office desk with a variety of objects"]

**A:** ["a cluttered home office desk", "a variety of objects on the desk"]

**Q:** ["a computer monitor is situated towards the left side of the desk, accompanied by a keyboard and a mouse placed directly in front of it"]

**A:** ["a computer monitor", "the left side of the desk", "a keyboard", "a mouse"]

Figure 8: LLM prompts for shorter text phrase $T$ extraction. "Q" is the example query sentence, and "A" is the expected shorter phrase output.

**Q:** ["there are several cars parked on the street, one of which is a red car near the crosswalk"]

**A:** ["there are several cars parked on the street", "a red car near the crosswalk"]

**Q:** ["on the countertop, there is a white plate and a bowl, two cups, a spoon, and a bottle"]

**A:** ["a white plate on the countertop", "a bowl on the countertop", "two cups on the countertop", "a spoon on the countertop", "a bottle on the countertop"]

**Q:** ["the image features a cluttered home office desk with a variety of objects"]

**A:** ["a cluttered home office desk", "a variety of objects on the office desk"]

**Q:** ["a computer monitor is situated towards the left side of the desk, accompanied by a keyboard and a mouse placed directly in front of it"]

**A:** ["a computer monitor is situated towards the left side of the desk", "a keyboard and a mouse placed directly in front of the monitor"]

Figure 9: LLM prompts for longer text phrase $T$ extraction. "Q" is the example query sentence, and "A" is the expected longer phrase output.

### A.3 IMAGE2TEXT AND CONCEPT2TEXT: TEXT PHRASE EXTRACTION

Unlike image descriptions obtained from *Concatenation* or *Text2Text* strategies, which consist of a list of phrases, the variants in *Image2Text* and *Concept2Text* are expressed as paragraphs. Due to the ineffectiveness of the "period" or "comma" segment (refer to Table 1), we experimented with partitioning the sentences by phrase extraction through an LLM. We randomly sample four sentences (*i.e.,* segmented by "period") and extract phrases manually as in-context examples. Fig. 8 presents examples of shorter phrases, while Fig. 9 shows examples of longer phrases.

## B SELECTION OF BASE MODEL

It is non-trivial to select a model that can extensively examine the quality of synthetic data for visual grounding. We select ALBEF (Li et al., 2021) as our base model due to its reported off-the-shelf visual grounding performance and success in further improvement with an attention mask consistency objective (Yang et al., 2023). Moreover, we intend to generate synthetic data that is effective for both weakly and box-supervised methods, such as the real VG data. The desired model is supposed to be

Table 9: VLM's off-the-shelf, weakly-supervised tuned, and AMC box-supervised tuned visual grounding performance in pointing game accuracy.

| Model | Box (AMC) | Data | RefCOCO+ | | Flickr30k | $\Delta_{avg}$ |
|-------|-----------|------|----------|---------|-----------|----------------|
| | | | Test A | Test B | | |
| CLIP | × | Off-the-Shelf | 47.42 | 41.36 | 59.22 | - |
| | × | VG | 44.38 | 39.09 | 54.95 | -3.19 |
| | ✓ | VG | 33.29 | 35.71 | 46.87 | -10.71 |
| METER | × | Off-the-Shelf | 68.07 | 52.73 | 83.16 | - |
| | × | VG | 53.44 | 34.73 | 57.65 | -19.38 |
| | ✓ | VG | 83.16 | 65.58 | 88.95 | +11.24 |
| BLIP | × | Off-the-Shelf | 58.56 | 38.00 | 64.54 | - |
| | × | VG | 68.86 | 52.85 | 64.08 | +8.23 |
| | ✓ | VG | 78.47 | 61.96 | 85.35 | +21.56 |

improved with and without box supervision, so that the investigation can be conducted continually and consistently from image-text synthesis to image-text-box synthesis.

To the best of our knowledge, ALBEF is the only VLM fine-tuned with a proposed box-supervised objective (AMC) achieving the current state-of-the-art. Hence, we make our best effort to extract gradient-based explanation maps from other VLMs and implement the AMC loss on top of them. As shown in Table 9, we explore 3 other models, CLIP (Radford et al., 2021), BLIP (Li et al., 2022a), and METER (Dou et al., 2022).

**CLIP.** We extract the GradCAM (Selvaraju et al., 2017) attention map from the last layer of the image encoder using its contrastive loss. The weakly-supervised training adopts its contrastive loss. We implement the AMC loss on top of it for box-supervised training. To our best effort, we can not obtain positive results from CLIP by tuning it on the real VG dataset, either weakly or fully. Therefore, CLIP is not a proper choice for visual grounding experiments.

**METER.** We pick the 5th layer of the cross-modal image encoder and obtain the GradCAM attention map from the image-text matching loss. The weakly-supervised experiments fine-tune METER with its original losses. The box-supervised experiments fine-tune METER with its original losses and an AMC loss we implemented. METER's original losses are deficient for weakly-supervised tuning. Using the original loss, our best effort for fine-tuning METER on VG significantly decreases performances. When using AMC loss instead, the real data improves METER by 11.24%. Given the deficiency under the weakly-supervised setting, METER can not be easily adopted to investigate both image-text and image-text-boxes continuously.

**BLIP.** We extract the GradCAM attention maps from the 8th layer of cross-modal attention from image-text matching loss. The weakly-supervised experiments finetune BLIP with its original losses. The box-supervised one fine-tune BLIP with its original losses and an AMC loss we implemented. Training on VG image-text pairs and image-text-box triplets both boosts the grounding performance. Therefore, we select BLIP as an additional model to verify the generalization of our findings.

## C  ABLATIONS AND FINDINGS WITH BLIP

Table 10 provides ablation studies with BLIP fine-tuned on synthetic image-text pairs. We observe consistent performance change as ALBEF's in Table 1. The Concatenation and Text2Text strategies are ineffective for BLIP as well. In the Image2Text strategy, the shorter phrases extracted from LLaVA captions also fit BLIP better. It is likely due to the nature of visual grounding that focuses on a small RoI (shorter phrases) instead of the entire image or broader RoIs (longer phrases). Also, the longer phrases defined by our prompts contain complex compositions which may affect VLM's performance. In Table 11, we fine-tune BLIP with image-text-box triplets. Longer phrases (row 3) result in less improvement than the shorter phrases (row 4), which consistently aligns with the findings from ALBEF.

Table 10: Comparisons of image-text synthesis strategies. We assess the effectiveness of synthetic image-text pairs from text concatenation, Text2Text, and Image2Text pipelines by evaluating the performance improvements over a BLIP model Li et al.. For reference we also include the performance that would be obtained by finetuning BLIP on real image-text pairs from Visual Genome (VG).

| Category | No. | Image | Text | Num. | RefCOCO+ | | Flickr30k | $\Delta_{avg}$ |
| | | | | | Test A | Test B | | |
| --- | --- | --- | --- | --- | --- | --- | --- | --- |
| BLIP | 1 | - | - | – | 58.56 | 38.00 | 64.54 | - |
| BLIP + VG | 2 | VG | VG | 1,649,546 | 68.86 | 52.85 | 64.08 | +8.23 |
| Concatenation | 3 | Syn-C | VG | 1,649,546 | 60.29 | 41.58 | 50.98 | -2.75 |
| Text2Text | 5 | Syn-V | $\text{LLM}_C$ | 530,233 | 61.79 | 42.42 | 55.09 | -0.60 |
| Image2Text | 6 | Syn-L | $\text{LLaVA}_L$ | 680,093 | 61.56 | 42.03 | 57.69 | +0.06 |
| | 7 | Syn-L | $\text{LLaVA}_S$ | 1,031,521 | **63.45** | **44.39** | **68.21** | **+4.98** |

Table 11: Effectiveness of synthetic image-text-boxes generated with GLIP (Li et al., 2022b). For reference we also include the performance that would be obtained by finetuning BLIP (Li et al., 2022a) with an AMC loss (Yang et al., 2023) on real image-text-box triplets from Visual Genome (VG).

| No. | Image | Text | Box | Num. | RefCOCO+ | | Flickr30k | $\Delta_{avg}$ |
| | | | | | Test A | Test B | | |
| --- | --- | --- | --- | --- | --- | --- | --- | --- |
| 1 | - | - | - | – | 58.56 | 38.00 | 64.54 | - |
| 2 | VG | VG | VG | 1,649,546 | 78.47 | 61.96 | 85.35 | +21.56 |
| 3 | Syn-L | $\text{LLaVA}_L$ | GLIP | 659,927 | 68.46 | 54.43 | 85.73 | +15.84 |
| 4 | Syn-L | $\text{LLaVA}_S$ | GLIP | 998,406 | **71.78** | **54.82** | **85.83** | **+17.11** |

# D  SYNTHETIC TEXT ANALYSIS

This section supplements the analysis of the factors causing the performance gap with the real data in Sec 3.5. Specifically, here we focus on analyzing the similarity, diversity, and coverage of synthetic text $T$ and real text $T^R$.

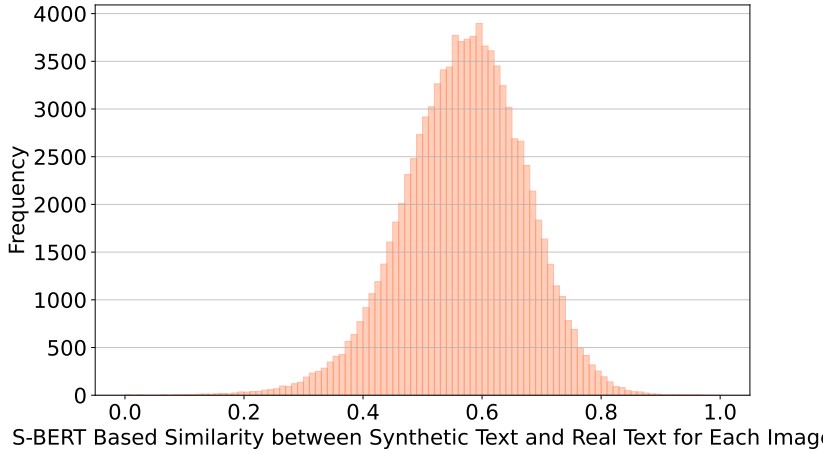

Figure 10: Distribution of image-wise average Sentence-BERT (Reimers & Gurevych, 2019) based cosine similarity between synthetic and real text.

To compute the text similarity, we adopt a pretrained Sentence-BERT (Reimers & Gurevych, 2019) to encode text into embeddings. Cosine similarity is then calculated between the embeddings of synthetic and real text corresponding to each image. We determine the text similarity for each

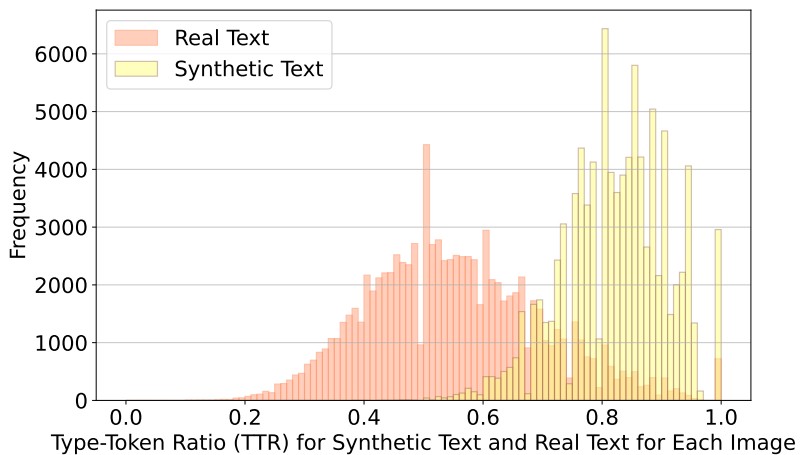

Figure 11: Distribution of image-wise type-token ratio for synthetic and real text.

synthetic caption by selecting the highest similarity among the real text embeddings, and then track the average similarity for each image. The distribution of the average similarity between synthetic and real text for each image is depicted in Fig. 10, where the highest frequency shows a score of around 0.6. The dissimilarity between the synthetic and real text aligns with the observation of degradation from text synthesis compared to real text (Table 4).

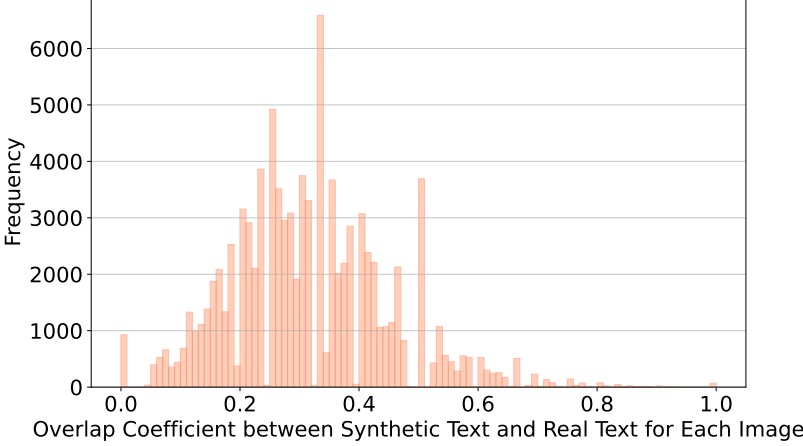

Figure 12: Distribution of image-wise overlap coefficient between synthetic and real text.

To delve deeper into the distinctions between synthetic and real texts, we compare their text diversity and coverage. Text diversity is measured using the Type-Token Ratio (TTR), which calculates the ratio of unique token types to the total number of tokens in a text. As shown in Fig. 11, our synthetic text $T$ generally has greater diversity than the real text $T^R$ from VG, indicating more elaborate descriptions that, however, may risk including irrelevant words to the visual content. Additionally, we calculate the overlap coefficient between the unique words in synthetic and real text, assessing the coverage and intersection of vocabulary, peaking at around 0.3 (See Fig. 12). This relatively low coefficient reveals the difference in word usage or content focus between the synthetic text $T$ and the real text $T^R$.

The observation of a higher TTR in synthetic texts $T$ with a modest overlap coefficient with real texts $T^R$ suggests a trade-off for synthesizing more effective texts for visual grounding. Although

the broader vocabulary in synthetic texts $T$ suggests richer and more diverse word usage as well as lower repetition when describing an image, the low overlap score implies a divergence from human-annotated content. Moreover, the presence of approximately 600K fewer texts in the synthetic data may indicate that paraphrasing in real data plays a crucial role.

# E    ALTERNATIVE PARADIGM DESIGN

Table. 13 compares two strategies of distinct sequence on phrase extraction (See Fig. 14). The "Caption" strategy adopted in SynGround obtains the synthetic phrases for visual grounding by applying LLM phrase extraction on captions derived from captioning the real VG images (row 2). Alternatively, "ReCaption" extracts phrases from paragraphs captioned on synthetic images. The core comparison between the "Caption" and "ReCaption" paradigms essentially boils down to evaluating the visual-textual misalignment introduced by image synthesis via a text-to-image model ("Caption") against the misalignment from an image captioning model ("ReCaption"). Table. 13 reveals a consistent observation with Table 4 that information loss or misalignment stems from the text synthesis, specifically image captioning on synthetic images in this experiment, rather than the image synthesis.

Figure 13: Data synthesis comparisons. "ReCaption" denotes applying an image captioning model to synthetic images, whereas "Caption" is applied on real images.

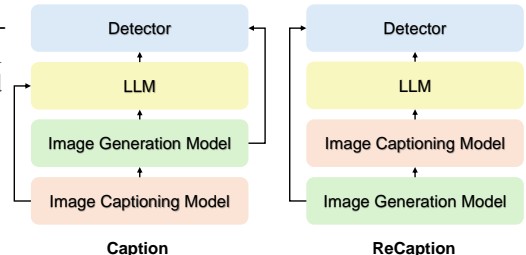

| Paradigm | RefCOCO+ | | Flickr30k |
|---|---|---|---|
| | Test A | Test B | |
| ReCaption | 73.09 | 56.33 | 86.80 |
| Caption | 73.70 | 56.35 | 86.89 |

Figure 14: Comparative overview of the "Caption" and "ReCaption" strategies.

# F    QUALITATIVE EXAMPLES

In this section, we supplement additional qualitative examples of our synthetic image-text-boxes. For better display, we randomly present a text phrase if there are multiple phrases for overlapping boxes (IoU $\geq$ 0.95). The full dataset will be released upon publication.

In Fig. 15, the first row showcases indoor scenes, the second row features human-related scenes, and the third row depicts outdoor scenes. Intriguingly, our synthetic data shows diversity, such as unconventional design (*e.g.,* "the lamp") or color (*e.g.,* "a green chair", "chairs with a floral pattern", "red pillows") of furniture in the first row. Despite the presence of artifacts, synthetic humans generally have human-like shapes (row 2). Considering the experimental results (refer to Table 3) that tuning on synthetic data improves grounding performance on the RefCOCO+ Test A, a person-only benchmark, the synthetic human with artifacts still benefits visual grounding. The third row presents some challenging scenarios, including small objects (e.g., "traffic lights," "a train"), detailed descriptions (e.g., "a well-maintained grassy yard"), and complex grammatical structures (e.g., "covered with snow"). In Fig. 16, similar properties are also found in synthetic data generated with less real data reliance (Sec 3.7). Overall, synthetic data with artifacts is able to improve visual grounding performance based on our result, but we expect learning from more advanced image-generative models or text-generative models can lead to further enhancement.

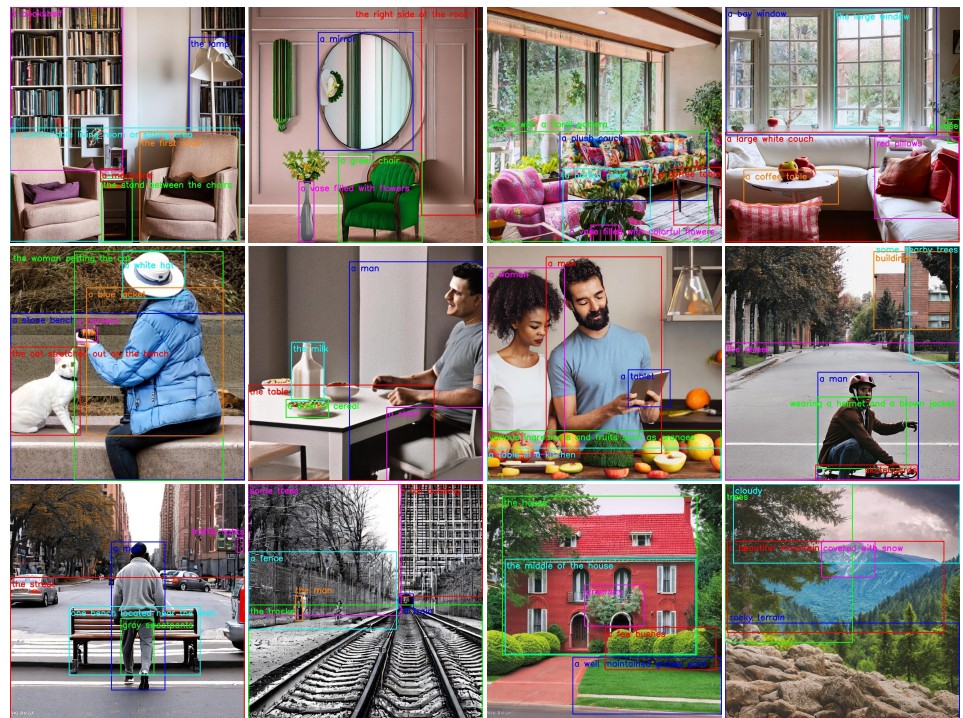

Figure 15: Qualitative examples of our synthetic image-text-boxes. The images are synthesized by a text-to-image generative model. The texts are generated by an LLM, and their corresponding boxes are obtained from an open-vocabulary object detector.

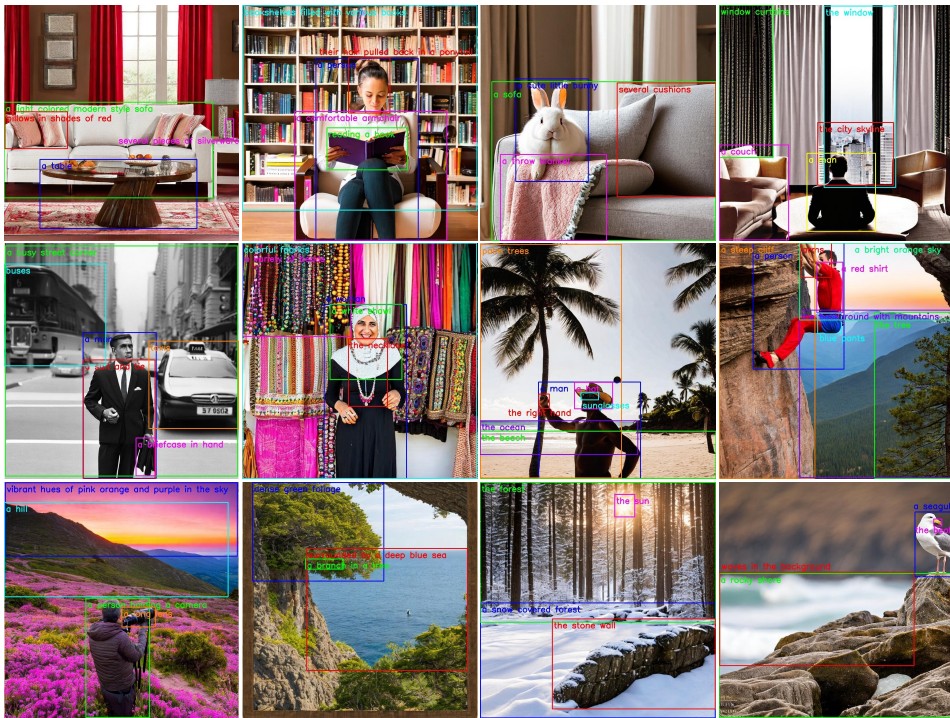

Figure 16: Qualitative examples of our synthetic image-text-boxes generated with *Concept2Text* strategy that relies less on the real data. The images are synthesized according to LLM-generated image descriptions through a text-to-image generative model. The texts are generated by an LLM, and their corresponding boxes are obtained from an open-vocabulary object detector.

