# OpenReview forum: "Is Synthetic Data Ready for Improving Visual Grounding?"
_ICLR.cc/2025/Conference — ICLR 2025 Conference Withdrawn Submission_

### Official Review · Reviewer_4VzW · 2024-10-18

**Soundness:** 2
**Presentation:** 2
**Contribution:** 2
**Rating:** 3
**Confidence:** 3

**Summary:**

This paper explores using synthetic data to improve visual grounding in vision-and-language models. The authors present SynGround, a pipeline that generates synthetic image-text-box triplets by combining advances in text-to-image generation, language models, and object detection. They compare synthetic data with real and web-crawled data on RefCOCO+ and Flickr30k benchmarks. Results show SynGround enhances localization in ALBEF and BLIP models, outperforming web-crawled data and offering potential for infinite data generation.

**Strengths:**

1. Systematic exploration: The paper systematically explores different strategies for generating synthetic image-text and image-text-box data, providing valuable insights into the factors influencing performance. The paper compares the performance of models trained on synthetic data with models trained on real and web-crawled data.

2. Pipeline for synthetic data generation: The proposed SynGround pipeline offers a structured approach for creating synthetic data for visual grounding, combining several advanced techniques.

3. Outperforming web-crawled data: The finding that synthetic data outperforms web-crawled data is a notable strength, suggesting the potential for creating more tailored and effective training datasets.

**Weaknesses:**

1. Use of older models: The paper relies on ALBEF and BLIP, which are relatively older models in the rapidly evolving field of vision and language. The performance in Experiment 1 does not compare to any of the models in the papersincode leaderboard (e.g., https://paperswithcode.com/sota/referring-expression-comprehension-on-refcoco-1). Evaluating SynGround with more recent and state-of-the-art models would significantly strengthen the claims.

2. Limited performance gains:  While improvements are reported, the absolute gains from using synthetic data, especially when combined with real data, are relatively modest and may not be statistically significant.  Error bars or further statistical analysis should be provided to support the claims of improvement.

3. Clarity and organization: The presentation of experiments could be improved.  The motivation and reasoning behind each experiment could be more clearly articulated.  Consolidating related experiments (like the BLIP experiments) into fewer tables would enhance readability.  The paper would benefit from focusing on the key findings, such as the comparison with web-crawled data, earlier in the presentation.

4. Lack of analysis on scaling limitations: While the paper mentions the potential for infinite data generation, it does not discuss or analyze potential limitations or saturation points in scaling up the use of synthetic data.

**Questions:**

Have you considered evaluating SynGround with more recent and state-of-the-art visual grounding models?

Could you elaborate on the computational resources required for generating and utilizing the synthetic data, especially in the context of scaling up to larger datasets?

Have you observed any limitations or saturation points when increasing the scale of synthetic data used for training?

Could you discuss the potential impact of biases present in the source data (e.g., caption descriptions) on the generated synthetic data and downstream visual grounding performance?

---

### Official Review · Reviewer_Jhhz · 2024-10-29

**Soundness:** 2
**Presentation:** 1
**Contribution:** 2
**Rating:** 3
**Confidence:** 3

**Summary:**

This paper proposes a pipeline that uses LLMs, object detector, and image generation model to improve the grounding ability of VLMs. They demonstrate that applying such a pipeline allows them to improve the performance of a baseline ALBEF model on grounding tasks (RefCOCO, Flickr30K).

**Strengths:**

- The ablations in the paper are rather comprehensive and highlight the importance of each part of the pipeline.
- The paper demonstrates that training on the synthetically generated data improves over the baseline results on grounding tasks.

**Weaknesses:**

- The baselines are weak: ALBEF is an older model that is far from SOTA on the benchmarks reported. What about applying the method to a more recent model (such as OFA [1])? One concern is that ALBEF is a much smaller model (BERT based LM), while the pipeline used to generate synthetic data leverage larger and more capable models such as LLaVA. I would be more convinced if the authors can apply their approach to improve a similar sized model. This would also alleviate concerns that this method is simply distilling a stronger ALBEF model from LLaVA generated data.
- The gains over the baselines are not really substantial enough at the moment to warrant running this (rather convoluted) synthetic generation pipeline. Even with the synthetic data, the relative improvements are worse than using real data (Table 2) and only marginally better when combined with real data (Table 3). In Figure 5, the improvement with introducing synthetic data also seems marginal, and within the error bounds of using less data (which does not bode well for scaling).
- The paper is rather difficult to read, and I found it structured in quite a confusing way. Figure 1 could be replaced with an overview of the full SynGround pipeline, including captioning, bounding box generation, image generation components, as well as the training objectives detailed in Sec 3.1.


**References**

[1] Wang, Peng, et al. "Ofa: Unifying architectures, tasks, and modalities through a simple sequence-to-sequence learning framework." International conference on machine learning. PMLR, 2022.

**Questions:**

- There are several versions of Llava (1, 1.5, 1.6) as well as different model sizes (7B, 13B, 34B). Which one is being used in this paper?
- Another common semi-synthetic pipeline is to re-caption images (e.g., see Sec 7.1.1 of [2]). How would such a recaptioning approach fare on the CC experiments in Sec. 3.9?

**References**

[2] Dubey, Abhimanyu, et al. "The llama 3 herd of models." arXiv preprint arXiv:2407.21783 (2024).

---

### Official Review · Reviewer_zEjc · 2024-11-04

**Soundness:** 2
**Presentation:** 3
**Contribution:** 3
**Rating:** 5
**Confidence:** 4

**Summary:**

This paper investigates the effectiveness of synthetic training data to improve the capabilities of vision-and-language models for grounding textual descriptions to image regions. They propose SynGround, an effective pipeline for generating useful synthetic data for visual grounding. Particularly, SynGround improves the pointing game accuracy of pretrained ALBEF and BLIP models by 4.81% and 17.11% absolute percentage points, respectively, across the RefCOCO+ and the Flickr30k benchmarks.

**Strengths:**

This paper provides a thorough experiments with different strategies to best generate image-text pairs and image-text-box triplets using a series of pretrained models under different settings and varying degrees of reliance on real data.

**Weaknesses:**

I really like the analysis in this paper. However, I'm confused about EFFECTIVENESS AND GENERALIZATION ON OTHER VLMS -- how general are the conclusions / findings in this paper (e.g. in Table 3 and 5), can they apply to more recent VLMs, since both the ALBEF and BLIP are smaller sized models from before 2022. Could you extend the method to more recent models such as LLaVA, Phi3.5, etc so that it is more likely to be a general conclusion?

**Questions:**

Happy to raise my score if weakness is addressed.

---

### Official Review · Reviewer_pC1Y · 2024-11-05

**Soundness:** 2
**Presentation:** 2
**Contribution:** 2
**Rating:** 3
**Confidence:** 3

**Summary:**

This paper studies a critical problem about data synthesis for improving visual grounding capabilities of vision and language models. It explores various strategies for generating synthetic image-text pairs and image-text-box triplets to enhance model training, comparing synthetic data with real and web-crawled data. The proposed SynGround pipeline demonstrates that synthetic data can effectively improve the localization capabilities of existing models. Notably, SynGround boosts pointing game accuracy for models like ALBEF and BLIP on benchmarks like RefCOCO+ and Flickr30k, showing the potential of synthetic data for scalable improvements in visual grounding tasks.

**Strengths:**

1. Visual grounding is an essential problem with current vision and language models. It's important to study an effective approach to build synthetic data to further scale up models' visual grounding capabilities. This paper is one of the approaches that study how to generate such data, and with comparisons of various approaches to generate such data.
2. SynGround improves the pointing game accuracy of pretrained ALBEF and BLIP significantly.

**Weaknesses:**

1. Previous synthetic visual grounding dataset are missing, for example, GRIT data - "a Ground-and-Refer Instruction-Tuning dataset with 1.1M samples.
GRIT contains multiple levels of spatial knowledge, covering objects, relationships, region descriptions, and complex reasoning" - proposed in Ferret is not compared with. It's not clear how proposed SynGround is differing from previous synthetic visual grounding data, and how it surpasses previous data generation approaches.
2. The main tables lack important SOTA baselines, for example, Shrika and Ferret on RefCOCO+ and Flickr, which are a lot better than the model fine-tuned on SynGround on RefCOCO+, and similar on Flickr.
3. In Table 1, also the proposed approach get average 0.36 marginal improvement, and also no better than directly fine-tuning on existing VG data, which get average 0.96 improvement.

**Questions:**

1. The selection of base vision and language models. Why not applied to more recent SOTAs. Is SynGround data benefiting recent SOTAs in visual grounding also?
2. How does SynGround compare with exisiting visual grounding data collected from public source with bounding boxes synthesized as well?

**Details Of Ethics Concerns:**

1. Discrimination / bias / fairness: Images with human-generated content may reflect biases, leading to fairness concerns in model training and predictions.
2. Legal compliance: Images containing identifiable human features may raise GDPR and copyright concerns if used without consent or proper authorization.
3. Responsible research: Releasing datasets with human-generated images requires careful handling to protect privacy and prevent potential misuse, especially if individuals are recognizable.

---

### Note · Authors · 2024-11-15

**Comment:**

We thank all the reviewers for their time and comments. We are withdrawing the paper from consideration and will address weaknesses in our next revision.

Here are some issues we are currently addressing including clarifications and experiments.

**Zero-Shot Out-of-the-Distribution Visual Grounding.** This paper adopts the zero-shot training setting that trains the visual grounding methods on collected image-text-boxes and evaluates it on task-specific benchmarks, such as RefCOCO+. The leaderboard and models such as “OFA” mentioned by the reviewers are finetuned on the in-domain training set (e.g., RefCOCO+ training split) and evaluated on its testing split. The in-domain training should achieve better performance, but we posit that the out-of-distribution data is more accessible and can examine the generalization ability of methods.

**Pointing Game Accuracy with Heatmaps vs. Accuracy with Boxes.** There are two standard settings for visual grounding, and they have different advantages. Heatmap visualizes the model’s attention, making grounding theoretically more closely where the models look. Not only is it more explainable, but it is more flexible compared to bounding boxes in terms of multiple objects or background regions. However, comparing the absolute value of pointing game accuracy with the Accuracy\@0.5 is unfeasible.

**Why AMC (ALBEF)?**  To the best of our knowledge,  AMC [CVPR 2023], which adopts ALBEF as the backbone model, is still the state-of-the-art zero-shot grounding method under the pointing game accuracy and without finetuning on the training split for individual downstream datasets. We provide several other backbones to verify the effectiveness of our synthetic data and the generalization of analysis. Additionally, we want to generate effective synthetic data for both unsupervised (image-text pairs) and supervised (image-text-box triplets) learning. It is non-trivial to select a backbone that can be used for both supervised and unsupervised training (See Appendix B).

**Finetuning SotA Box-based Acc\@0.5 Model (OFA).** Our synthetic data can improve both heatmap-based and box-based grounding methods. The Acc\@0.5 SotA at RefCOCO+, OFA, mentioned by the reviewers, was trained on the RefCOCO+ training split and tested on the RefCOCO+ testing split. Here, we evaluated the zero-shot grounding performance on RefCOCO+ and finetuned it for out-of-distribution zero-shot performance. The off-the-shelf OFA-Base without finetuning on RefCOCO+ is much lower than the in-domain (row 1) training-testing result. However, our synthetic data improves OFA dramatically and comes close to VG finetuning.

|  | Finetuning Data | RefCOCO+ Val   | RefCOCO+ Test A | RefCOCO+ Test B |
|-----------------|----------|-------|--------|--------|
| OFA_Base        | RefCOCO+ | 81.39 | 87.15  | 74.29  |
| OFA_Base        | -        | 29.78 | 31.24  | 27.82  |
| OFA_Base        | VG       | 54.29 | 59.52  | 48.19  |
| OFA_Base        | SynGround| 49.53 | 52.31  | 45.37  |

**LLaVA for Visual Grounding.** LLaVA does not provide a downstream application for visual grounding, and there is no straightforward approach to using LLaVA for this purpose. Unlike VLMs that use image-text matching or contrastive loss for attention map extraction, LLaVA is trained with an autoregressive loss, making it unclear how to extract a GradCAM explanation. Adapting LLaVA for visual grounding would require significant modifications, such as integrating an additional box decoder or adding location tokens during training, which are beyond the scope of our research on data synthesis.

Notably, the LLaVA model adopts the CLIP image encoder, the same as ALBEF, BLIP, and METER. By experimenting with ALBEF, BLIP, and METER, we demonstrate the effectiveness of our synthetic data through extensive experiments, potentially indicating that grounding improvements could be achieved for LLaVA if its structure is modified to suit the grounding task.


**Computation Costs.** The data scale of our synthetic data is at approximately 100K images and 1M text-box pairs. Below are the computation speeds tested on a single NVIDIA A40 GPU. The entire image-text-box synthesis takes 501 hours = 20.88 days on a single card.
- Image caption generation (LLaVA): 5.71s/it *100K = 158 GPU hours.
- Image synthesis (Stable diffusion): 4.85s/it * 100K = 135 GPU hours.
- Text synthesis (LLM): 0.52s/it * 1M = 144 GPU hours.
- Box synthesis (GLIP): 0.23s/it * 1M = 64 GPU hours.

Quote from VG paper [3]: The dataset was curated with contributions from over 33,000 unique workers over six months, following 15 months of experimentation and refinement of data representation.

**Effectiveness at the Same Efforts: Synthetic Data vs. Real Data.** Compared to the image-text pairs, image-text-box triplets are more laborious to curate. The scale for the existing image-text-box dataset is much smaller than the image-text datasets (e.g., LAION5B).
Within the same data curation (collection & annotation) time period, SynGround’s 20.88 GPU days are 1/9 of VG’s data curation time from 33,000 unique workers. Below, we provide comparisons between our synthetic data and 1/9 VG data. Their performance is on par. Plus, the scaling up observed in Sec. 3.8. Except for the analysis and findings from the data synthesis in our paper, SynGround provides a potentially feasible way to curate image-text-boxes at scale.

 |          | Data  | RefCOCO+ Test A | RefCOCO+ Test B |  Flickr30k |  Avg Δ|
|--------------|----------|-----------|-------|--------|--------|
| Off-the-Shelf| -        | 69.35     | 53.77 | 79.38  | -      |
| SynGround    | Synthetic| 73.70     | 56.35 | 86.89  | +4.81  |
| 1/9 VG       | Real     | 76.96     | 59.07 | 85.01  | +6.18  |

**Withdrawal Confirmation:**

I have read and agree with the venue's withdrawal policy on behalf of myself and my co-authors.